# NH Stretching Frequencies of Intramolecularly Hydrogen-Bonded Systems: An Experimental and Theoretical Study

**DOI:** 10.3390/molecules26247651

**Published:** 2021-12-17

**Authors:** Poul Erik Hansen, Mohammad Vakili, Fadhil S. Kamounah, Jens Spanget-Larsen

**Affiliations:** 1Department of Science and Environment, Roskilde University, Universitetsvej 1, DK-4000 Roskilde, Denmark; 2Department of Chemistry, Faculty of Science, Ferdowsi University of Mashhad, Mashhad 91775-1436, Iran; vakili-m@um.ac.ir; 3Department of Chemistry, University of Copenhagen, Universitetsparken 5, DK-2100 Copenhagen, Denmark; fadil@chem.ku.dk

**Keywords:** secondary amines, NH stretching wavenumbers, NH/ND isotopic wavenumber ratios, NMR chemical shifts, DFT calculations, anharmonicity

## Abstract

The vibrational NH stretching transitions in secondary amines with intramolecular NH···O hydrogen bonds were investigated by experimental and theoretical methods, considering a large number of compounds and covering a wide range of stretching wavenumbers. The assignment of the NH stretching transitions in the experimental IR spectra was, in several instances, supported by measurement of the corresponding ND wavenumbers and by correlation with the observed NH proton chemical shifts. The observed wavenumbers were correlated with theoretical wavenumbers predicted with B3LYP density functional theory, using the basis sets 6-311++G(d,p) and 6-31G(d) and considering the harmonic as well as the anharmonic VPT2 approximation. Excellent correlations were established between observed wavenumbers and calculated harmonic values. However, the correlations were non-linear, in contrast to the results of previous investigations of the corresponding OH···O systems. The anharmonic VPT2 wavenumbers were found to be linearly related to the corresponding harmonic values. The results provide correlation equations for the prediction of NH stretching bands on the basis of standard B3LYP/6-311++G(d,p) and B3LYP/6-31G(d) harmonic analyses, with standard deviations close to 38 cm^−1^. This is significant because the full anharmonic VPT2 analysis tends to be impractical for large molecules, requiring orders of magnitude more computing time than the harmonic analysis.

## 1. Introduction

It is well known that hydrogen-bonded linkages play major roles in a variety of areas in chemistry and molecular biology, and the investigation of such linkages by spectroscopic procedures is a subject of current interest [1]. Historically, IR absorption spectroscopy has been the most important spectroscopic tool in the investigation of hydrogen-bonded systems, typically involving OH···O and NH···O linkages [2,3,4]. The corresponding OH and NH stretching frequencies are thus indicative of the hydrogen bond strength [5]. However, the assignment of these bands in the measured IR spectra is frequently difficult because of overlap with signals due to traces of water or CH stretching vibrations. In systems with strong hydrogen bonds, the situation is further complicated by the influence of anharmonic effects, leading to band broadening and distribution of the intensity associated with the OH or NH stretching motion over several vibrational modes. In many cases, a specific OH or NH stretching band may be difficult to identify in the experimental IR spectrum [6,7,8,9,10].

The possibility to predict the vibrational transitions of hydrogen-bonded systems by theoretical methods has been of interest for decades [11]. Today, advanced procedures such as coupled-cluster (CCSD(T)) calculations can predict vibrational wavenumbers for small molecules within 10 cm^−1^ of the experimental values [12]. However, this level of theory is not feasible for large molecules, and in general, approximate methods must be applied. Most calculations of vibrational frequencies are performed within the harmonic approximation [13,14]. Scott and Radom [14] and Wong [15] investigated the harmonic vibrational frequencies obtained by a variety of calculational procedures for a large test set of molecular vibrations. They found that density functional theories (DFT) [13,16] such as B3LYP [17,18] and B3PW91 [17,19] led to the most successful correlations between observed and calculated vibrational wavenumbers. With a modest basis set such as 6-31G(d) and using a semiempirical scale factor close to 0.96, the application of these procedures resulted in an overall root-mean-square (RMS) error equal to 34 cm^−1^ [14,15]. Harmonic wavenumbers calculated by DFT procedures (in particular B3LYP) have thus been used to support the vibrational assignments in numerous IR spectroscopic investigations.

However, as indicated above, the harmonic approximation tends to fail for strongly hydrogen-bonded systems. A study was performed some time ago of a series of OH···O systems including weak, strong, and very strong intramolecular hydrogen bonds, with OH stretching wavenumbers ranging from 3600 to 1900 cm^−1^ [20,21]. Standard B3LYP harmonic analyses with the usual scale factors led to the prediction of too large OH stretching wavenumbers for systems with strong and short hydrogen bonds, evidently due to lowering of the wavenumbers by anharmonic effects. On the other hand, application of the VPT2 anharmonic approximation developed by Barone et al. [22,23] overestimated the anharmonic lowering of the OH stretching wavenumbers, leading to predictions of much too low wavenumbers for strong hydrogen bonds, for example, almost 900 cm^−1^ too low for nitromalonamide enol with B3LYP/6-31G(d) and more than 1000 cm^−1^ too low for dibenzoylmethane enol with B3LYP/cc-pVDZ [20,21,24,25]. Fortunately, a satisfactory linear correlation was established between observed OH stretching wavenumbers and calculated harmonic values in the complete experimental range from 3600 to 1900 cm^−1^, thereby enabling a convenient prediction of effective OH stretching band centers from the results of a standard harmonic analysis [20,21].

**Scheme 1 molecules-26-07651-sch001:**
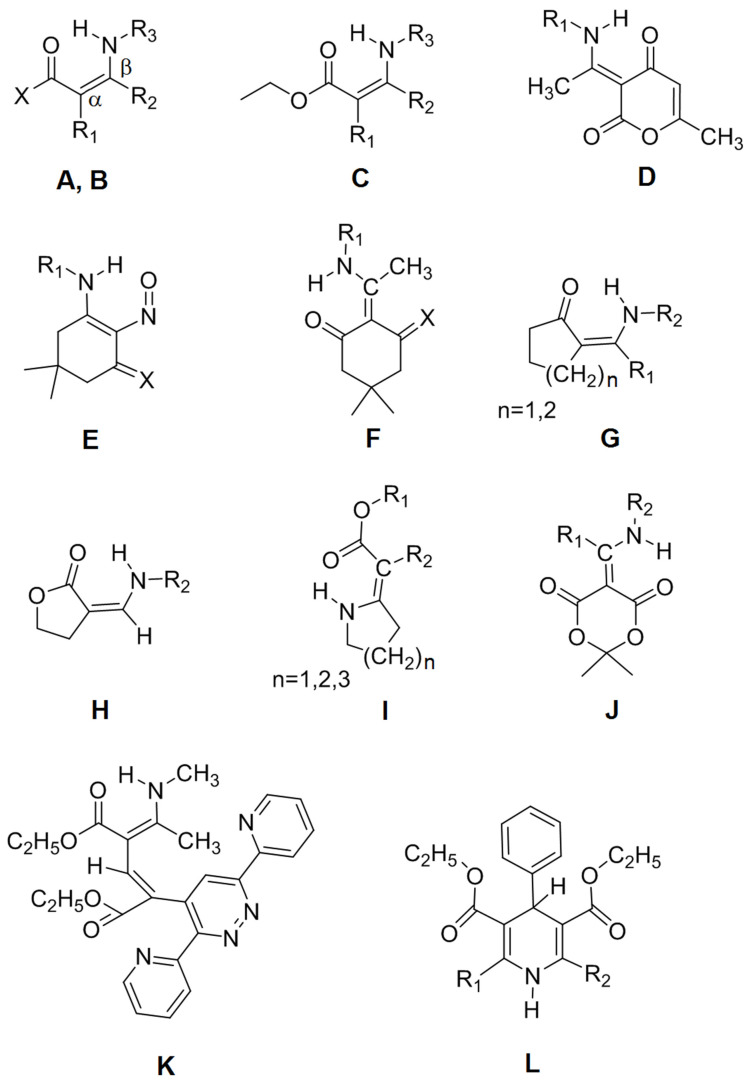
General structures of the investigated compounds. For substituents and names, see Table 1.

**Table 1 molecules-26-07651-t001:** The compounds investigated in this study (for type, etc., see Figure 1).

No.	Type	R_1_	R_2_	R_3_	X	Name
**1**	A	H	H	*t*-Butyl	H	(*Z*)-3-(tert-butylamino)acrylaldehyde
**2**	A	H	H	Ethyl	CH_3_	(*Z*)-4-(ethylamino)but-3-en-2-one
**3**	A	H	H	Ethyl	CF_3_	(*Z*)-4-(ethylamino)-1,1,1-trifluorobut-3-en-2-one
**4**	A	CH_3_	H	CH_3_	CF_3_	(*Z*)-1,1,1-trifluoro-3-methyl-4-(methylamino)but-3-en-2-one
**5**	A	H	NO_2_	CH_3_	CH_3_	(*E*)-4-methylamino-4-nitrobut-3-en-2-one
**6**	B	H	CH_3_	Ph (phenyl)	CH_3_	(*Z*)-4-(phenylamino)pent-3-en-2-one
**7**	B	H	CH_3_	*p*-ClPh	Ph	(*Z*)-3-(4-chlorophenylamino)-1-phenylbut-2-en-1-one
**8**	B	H	CH_3_	*o*-CH_3_Ph	CH_3_	(*Z*)-4-(o-tolylamino)pent-3-en-2-one
**9**	B	H	CH_3_	*o*-FPh	CH_3_	(*Z*)-4-(2-fluorophenylamino)pent-3-en-2-one
**10**	B	H	CH_3_	*o*-ClPh	CH_3_	(*Z*)-4-(2-chlorophenylamino)pent-3-en-2-one
**11**	B	H	CH_3_	2,6-diCH_3_Ph	CH_3_	(*Z*)-4-(2,6-dimethylphenylamino)pent-3-en-2-one
**12**	B	H	CH_3_	CH_3_	CH_3_	(*Z*)-4-(methylamino)pent-3-en-2-one
**13**	B	H	Ph	Isopropyl	CF_3_	(*Z*)-1,1,1-trifluoro-4-(isopropylamino)-4-phenylbut-3-en-2-one
**14**	B	CH_3_	CH_3_	CH_3_	CH_3_	(*Z*)-3-methyl-4-(methylamino)pent-3-en-2-one
**15**	B	CH_3_	CH_3_	*p*-OCH_3_Ph	CH_3_	(*Z*)-4-(4-methoxyphenylamino)-3-methylpent-3-en-2-one
**16**	C	H	CH_3_	Ph	-	(*Z*)-ethyl 3-(phenylamino)but-2-enoate
**17**	C	H	CH_3_	CH_3_	-	(*Z*)-ethyl 3-(methylamino)but-2-enoate
**18**	C	CH_3_	CH_3_	Ph	-	(*Z*)-ethyl 2-methyl-3-(phenylamino)but-2-enoate
**19**	C	CH_3_	CH_3_	CH_3_	-	(*Z*)-ethyl 2-methyl-3-(methylamino)but-2-enoate
**20**	C	CH_3_	CH_3_	CH_2_Ph	-	(*Z*)-ethyl 3-(benzylamino)-2-methylbut-2-enoate
**21**	C	H	CH_3_	CH_2_Ph	-	(*Z*)-ethyl 3-(benzylamino)but-2-enoate
**22**	D	CH_2_CH_2_NH_2_	-	-	-	(*E*)-3-(1-(2-aminoethylamino)ethylidene)-6-methyl-2*H*-pyran-2,4(3*H*)-dione
**23**	D	*p*-CH_3_OPh	-	-	-	(*E*)-3-(1-(4-methoxyphenylamino)ethylidene)-6-methyl-2*H*-pyran-2,4(3*H*)-dione
**24**	D	*p*-ClPh	-	-	-	(*E*)-3-(1-(4-chlorophenylamino)ethylidene)-6-methyl-2*H*-pyran-2,4(3*H*)-dione
**25**	E	*m*-CH_3_OPh	-	-	-	5,5-dimethyl-3-(*m*-anisidino)-2-nitroso-2-cyclohexen-1-one
**26**	E	Ph	-	-	-	5,5-dimethyl-3-anilino-2-nitroso-2-cyclohexen-1-one
**27**	F	CH_3_	-	-	-	5,5-dimethyl-2-(1-(methylamino)ethylidene)cyclohexane-1,3-dione
**28**	F	Iso-propyl	-	-	-	2-(1-(isopropylamino)ethylidene)-5,5-dimethylcyclohexane-1,3-dione
**29**	F	Ph	-	-	-	5,5-dimethyl-2-(1-(phenylamino)ethylidene)cyclohexane-1,3-dione
**30**	G	H	CH_3_	-	*n* = 1	(*Z*)-2-((methylamino)methylene)cyclopentanone
**31**	G	CH_3_	CH_3_	-	*n* = 1	(*Z*)-2-(1-(methylamino)ethylidene)cyclopentanone
**32**	G	H	Bu	-	*n* = 2	(*Z*)-2-((butylamino)methylene)cyclohexanone
**33**	H	CH_3_	-	-	-	(*E*)-3-((methylamino)methylene)dihydrofuran-2(3*H*)-one
**34**	H	Ph	-	-	-	(*Z*)-3-((phenylamino)methylene)dihydrofuran-2(3*H*)-one
**35**	I	Et	COOEt	-	*n* = 1	diethyl 2-(pyrrolidin-2-ylidene)malonate
**36**	I	Et	COOEt	-	*n* = 2	diethyl 2-(piperidin-2-ylidene)malonate
**37**	I	Et	COOEt	-	*n* = 3	diethyl 2-(azepan-2-ylidene)malonate
**38**	J	CH_3_	CH_3_	-	-	2,2-dimethyl-5-(1-(methylamino)ethylidene)-1,3-dioxane-4,6-dione
**39**	J	Et	CH_2_COOEt	-	-	ethyl 2-(1-(2,2-dimethyl-4,6-dioxo-1,3-dioxan-5-ylidene)propylamino)acetate
**40**	K	-	-	-	-	(2*E*,4*Z*)-diethyl 2-(3,6-di(pyridin-2-yl)pyridazin-4-yl)-4-(1-(methylamino)-ethylidene)pent-2-enedioate
**41**	L	CH_2_OCOCH_3_	CH_2_OCOCH_3_	-	-	diethyl 2,6-bis(acetoxymethyl)-4-phenyl-1,4-dihydropyridine-3,5-dicarboxylate
**42**	L	CH_3_	CH_2_OCOCH_3_	-	-	diethyl 2-(acetoxymethyl)-6-methyl-4-phenyl-1,4-dihydropyridine-3,5-dicarboxylate
**43**	L	CH_3_	CH_3_	-	-	diethyl 2,6-dimethyl-4-phenyl-1,4-dihydropyridine-3,5-dicarboxylate
**44**	B	H	CH_3_	*p*-PhCOOEt	CH_3_	(*Z*)-ethyl 4-(4-oxopent-2-en-2-ylamino)benzoate
**45**	B	CH_3_	CH_3_	Ph	CH_3_	(*Z*)-3-methyl-4-(phenylamino)pent-3-en-2-one

**Table 2 molecules-26-07651-t002:** Theoretical NH stretching wavenumbers (cm^−1^; Harm = harmonic, Anh = VPT2 anharmonic) and NH bond lengths R_NH_ (Å) computed with B3LYP. Observed or estimated wavenumbers ν_NH_ (cm^−1^) and observed NH proton chemical shifts δ_NH_ (ppm). Entries in italics indicate wavenumbers predicted by Equations (1) and (2).

Compound ^1^	6-31G(d)				6-311++G(d,p)		Observed		
	R_NH_	Harm	Anh	*P(Harm)* ^2^	Harm	*P(Harm)* ^3^	ν_NH_	δ_NH_	Ref.
**1**	1.0236	3410	3208	*3247*	3424	*3251*	3195	-	[26]
**2**	1.0224	3404	3176	*3242*	3400	*3232*	3190	-	[27]
**3**	1.0238	3411	3208	*3248*	3422	*3250*	3222	-	[28]
**4**	1.0221	3432	3214	*3265*	3441	*3264*	3206	10.21	[29]
**5**	1.0270	3341	3079	*3182*	3362	*3200*	3180	-	[30]
**6**	1.0303	3274	2932	*3101*	3244	*3071*	3031 ^4,5^	12.48	[31]
**7**	1.0315	3209	2909	*3001*	3242	*3068*	3056	13.07	[32]
**8**	1.0305	3266	2973	*3090*	3253	*3082*	3058 ^5^	12.34	[33]
**9**	1.0307	3268	2943	*3093*	3258	*3089*	3063 ^5^	12.25	[33]
**10**	1.0306	3276	-	*3104*	3260	*3091*	3047 ^5^	12.42	[33]
**11**	1.0311	3266	-	*3090*	3274	*3108*	3058 ^5^	11.95	[33]
**12**	1.0263	3342	3052	*3183*	3349	*3188*	3171	10.70	[9]
**13**	1.0272	3340	-	*3181*	3344	*3183*	3205	11.11	[34]
**14**	1.0265	3322	-	*3161*	3311	*3149*	3041 ^5^	11.86	-
**15**	1.0322	3226	-	*3030*	3205	*3017*	3004 ^4^	-	[35]
**16**	1.0244	3382	3147	*3223*	3388	*3223*	3254 ^4^	10.39	[7]
**17**	1.0194	3432	3247	*3265*	3469	*3283*	3295	8.46	[7]
**18**	1.0248	3363	-	*3205*	3360	*3198*	-	-	-
**19**	1.0199	3442	-	*3272*	3439	*3263*	3262	9.14	-
**20**	1.0215	3425	-	*3259*	3422	*3250*	3282	9.66 ^6^	[36]
**21**	1.0217	3427	-	*3261*	3442	*3265*	3289 ^7^	8.95	[37]
**22**	1.0377	3138	2782	*2860*	3102	*2843*	2870	14.18	[38]
**23**	1.0422	3059	2574	*2653*	3022	*2665*	2610	15.60	[38]
**24**	1.0440	3027	2591	*2551*	2978	*2548*	2602	15.90	[38]
**25**	1.0424	3050	2574	*2626*	3048	*2728*	2560 ^8^	18.41	[38]
**26**	1.0418	3048	2584	*2619*	3031	*2687*	2340 ^8^	18.35	[38]
**27**	1.0330	3208	2817	*2999*	3181	*2981*	-	13.3	[7]
**28**	1.0345	3172	2814	*2932*	3140	*2913*	2900 ^9^	13.6	[7]
**29**	1.0386	3124	2743	*2828*	3077	*2792*	2760	15.2	[7]
**30**	1.0231	3420	-	*3255*	3440	*3263*	-	8.84	[39]
**31**	1.0245	3367	-	*3209*	3374	*3210*	-	10.26	[40,41]
**32**	1.0253	3381	3121	*3222*	3390	*3224*	3160	-	[27]
**33**	1.0118	3599	-	*3359*	3613	*3362*	3302	6.60	[42]
**34**	1.0216	3449	-	*3277*	3466	*3281*	3314	9.05	[42]
**35**	1.0194	3481	-	*3299*	3488	*3296*	3317	9.52	[42]
**36**	1.0236	3385	-	*3226*	3372	*3209*	3242	10.08	[42]
**37**	1.0222	3405	-	*3243*	3411	*3241*	3280	8.83	[43]
**38**	1.0262	3341	-	*3182*	3331	*3170*	3224	11.32	[44]
**39**	1.0292	3318	-	*3157*	3304	*3142*	3172	11.60	[44]
**40**	1.0217	3415	-	*3251*	3414	*3244*	3270	-	[45]
**41**	1.0148	3558	-	*3341*	3559	*3336*	3403	7.7	[46]
**42**	1.0153	3541	-	*3333*	3551	*3332*	3347	6.6	[46]
**43**	1.0092	3631	-	*3372*	3636	*3371*	3336	5.6	[46]
**44**	1.0308	3266	-	*3090*	3237	*3062*	3047 ^5^	12.64	[47]
**45**	1.0322	3226	-	*3030*	3205	*3017*	2975 ^5^	13.46	[48]

^1^ See Table 1. ^2^ Wavenumbers predicted by Equation (2). ^3^ Wavenumbers predicted by Equation (1). ^4^ Not assigned as NH stretching in the referenced paper. ^5^ Estimated values (Table 3), not included in the correlation analyses. ^6^ NMR data from Ref. [36]. ^7^ Value varies slightly with solvent. ^8^ Not included in the correlation analyses, see Section 3.1. ^9^ Approximate.

**Table 3 molecules-26-07651-t003:** NH and ND stretching wavenumbers, ν_NH_ and ν_ND_ (cm^−1^), measured in CCl_4_ solution. Several ν_NH_ values are estimated on the basis of the ν_NH_/ν_ND_ ratio (footnote 2).

Compound ^1^	ν_NH_	ν_ND_
**6**	3031 ^2^	2262
**8**	3058 ^2^	2280
**9**	3063 ^2^	2286
**10**	3047 ^2^	2274
**11**	3058 ^2^	2282
**14**	3041 ^2^	2270
**16**	3254, 3185 ^3^	2403
**17**	3295	2431
**19**	3262	2411
**20**	3259	2409
**21**	3289	2435
**44**	3047 ^2^	2274 ^4^
**45**	2975 ^2^	2220

^1^ See Table 1. ^2^ Estimated from the observed ν_ND_, assuming ν_NH_/ν_ND_ = 1.34 (see Section 4). ^3^ Two bands are observed [7]; the larger wavenumber is used in the correlation analyses. ^4^ Ref. [48].

In this publication we present the results of a similar investigation of a large number of secondary amines with intramolecular NH···O hydrogen-bonded linkages. This type of linkage has the advantage that a wide range of compounds can be investigated because, in general, systems of the type NH···O=C with an intervening double bond are not tautomeric, in contrast to the corresponding OH···O=C systems. In those cases where the compounds are tautomeric, the data are not included. The structures and names of the investigated compounds are listed in Figure 1 and Table 1. The emphasis of our study is on the NH stretching bands of the NH···O linkages, covering wavenumbers in the range 3300–2600 cm^−1^. The aim is to develop simple correlation tools for the prediction of effective NH stretching wavenumbers for NH···O systems, to be useful in those cases where the NH stretching bands are not easily identified in the experimental spectra.

For the purpose of this investigation, a number of substances were synthesized and measured, but most spectroscopic data are quoted from the literature (Table 2). Vibrational assignments were in several cases supported by the investigation of deuterated compounds (NH hydrogen exchanged with deuterium). Further support was obtained by considering the correlation of NH stretching wavenumbers with ^1^H amine chemical shifts, following the original suggestion by Dudek [7]. Calculations of vibrational transitions were carried out with the B3LYP functional, using two different basis sets, and were performed both in the standard harmonic approximation and in the anharmonic VPT2 approximation. As we shall show, excellent correlations between observed and calculated NH stretching wavenumbers are obtained, but in contrast to the previous results for OH···O systems [20,21], the correlations are non-linear. Additional information is provided as Appendix A, referred to in the ensuing text as Appendix A.

## 2. Materials and Methods

### 2.1. Materials

The following substances were synthesized: **6**, **14**, **16**, **17**, **19**, and **45**. For details and characterization, see Appendix A. Deuterated compounds were prepared by dissolving the substances in CCl_4_ with a drop of trimethylamine [48]. After the addition of D_2_O (1 cm^3^), the mixture was stirred for 24 h, and the organic layer was separated and dried using Na_2_SO_4_.

### 2.2. Spectroscopy

The IR spectra of **6**, **8**, **9**, **10**, **11**, **14**, **16**, **17**, **19**, **44**, and **45** and of their deuterated analogs were recorded in the 4000–400 cm^−1^ region with a spectral resolution of 2 cm^−1^ by averaging the results of 10 scans on a PerkinElmer Spectrum 2000 FTIR spectrophotometer (Appendix A). The compounds were measured in KBr tablets and when possible also in CCl_4_ solutions. The CCl_4_ solutions were dried with Na_2_SO_4_ to remove traces of water from the samples. An example of the recorded spectra is shown in Figure 1, which displays the absorbance curves for normal and deuterated (*Z*)-ethyl 3-(methylamino)but-2-enoate (**17**) in the range 3800 to 900 cm^−1^. NMR spectra were recorded on Bruker Ultrashield Plus 500 MHz and Bruker Avance 3 spectrometers using CDCl_3_ as a solvent. For details and spectra, see Appendix A.

### 2.3. Calculational Details

Quantum chemical calculations were performed with the Gaussian 09 [49] and the Gaussian 16 [50] software packages producing similar results. Geometry optimizations and standard harmonic analyses were carried out in the gas phase with B3LYP density functional theory (DFT) [17,18] using the basis sets 6-311++G(d,p) and 6-31G(d) [13,49,50]. Several calculations using 6-31G(d) were carried out also with the anharmonic VPT2 approximation [22,23] (freq = anharmonic). The computed NH stretching wavenumbers for the compounds **1**–**45** are listed in Table 2. Detailed listings of results for **17** are provided as Appendix A, considering fundamentals in the range 800–3500 cm^−1^. For a selection of compounds (**1**, **6**, **16**, **17**, **23**, **29**, and **32**) covering a broad range of NH stretching wavenumbers, additional calculations were performed by using the B3PW91 functional [17,19]. Hydrogen bond energies were estimated using the AIM 2000 software at the B3LYP/6-31G(d) level [51] (Appendix A).

## 3. Results

### 3.1. Experimental NH Stretching Wavenumbers

Measured or estimated NH stretching wavenumbers for the compounds **1**–**45** are listed in Table 2 and Table 3. A subset of these compounds, the enaminones, has been thoroughly investigated by Gilli et al. [38]. NH stretching wavenumbers for the compounds **6, 8, 9, 11, 14, 44,** and **45** were estimated by measuring the IR spectra of the deuterated species and multiplying the observed ND stretching wavenumber with 1.34 (Table 3; see the discussion in Section 4). In these compounds, the NH stretching band appears to be overlapped by the CH stretching bands. The approximate wavenumbers derived in this manner are not included in the correlation analyses.

As a check of the assignments, the experimental NH stretching frequencies are correlated with the observed NH chemical shifts listed in Table 2. The available 26 data points are shown in Figure 2. A non-linear correlation is evident, in contrast to the linear relationships established in the literature, in particular by Dudek [7]. The points for the nitroso compounds **25** and **26** are outliers; they are not included in the correlation analysis (see Section 4). To a good approximation, the remaining 24 points follow an exponential function, leading to RMS standard deviation (SD) = 34.3 cm^−1^ (Figure 2). A similar fit of the data is obtained with a second-order polynomial function, but the exponential function is preferred because of a more satisfactory asymptotical behavior. The correlation equation in Figure 2 can be used to predict NH stretching wavenumbers from the observed chemical shifts.

### 3.2. Calculations

As pointed out in the Introduction Section, DFT calculations with the B3LYP functional are frequently used to support vibrational assignments in IR spectroscopic investigations. An example is given in Figure 3, which shows the scaling correlation of observed vibrational wavenumbers for the ester **17** and harmonic wavenumbers calculated with B3LYP/6-31++G(d,p), considering all fundamental wavenumbers in the range 800–3500 cm^−1^ (see Appendix A for details). A good fit is obtained, even for the NH stretching vibration observed close to 3300 cm^−1^ (Figure 1, Table 2). The intramolecular hydrogen bond in **17** is weak (Appendix A) and particular anharmonic effects are apparently of minor significance. However, when systems with stronger NH···O hydrogen-bonding are considered, the importance of anharmonic effects increases. Mroginski et al. [52] studied compounds with strong NH···N hydrogen bonds and found that separate scaling of the calculated NH force constants was required in order to reproduce the observed IR bands. Figure 4 shows the correlation of observed NH stretching wavenumbers and calculated harmonic values for the present NH···O linkages, yielding a non-linear correlation that strongly deviates from the linear one in Figure 3. The exponential correlation equation obtained on the basis of the data listed in Table 2 is given in Equation (1):B3LYP/6-311++G(d,p):
*P*(*Harm*) = 3509 − 6.310 × 10^6^·exp(−Harm/338.8)(1)
(32 points: R^2^ = 0.964, SD = 38.2 cm^−1^)

Harm is the calculated harmonic wavenumber, and P(Harm) is the value predicted with Equation (1). An SD close to 38 cm^−1^ seems quite acceptable, in view of the uncertainty of many of the experimental wavenumbers and the influence of different environmental effects. Results obtained with the much smaller basis set 6-31G(d) lead to a very similar correlation (Appendix A); the exponential correlation equation is given in Equation (2) as follows:B3LYP/6-31G(d):
*P*(*Harm*) = 3470 − 6.714 × 10^7^·exp(−Harm/270.3)(2)
(32 points: R^2^ = 0.965, SD = 38.1 cm^−1^)

In fact, the wavenumbers obtained with B3LYP/6-31G(d) and B3LYP/6-311++G(d,p) are linearly related with R^2^ = 0.991, SD = 15 cm^−1^ (S13). For predictive purposes, the cheaper B3LYP/6-31G(d) procedure is just as efficient as B3LYP/6–311++G(d,p).

The non-linearity of the present correlations is unexpected in view of the previous results for OH···O systems where linear relationships between observed and calculated OH stretching wavenumbers were obtained [20,21]. We carried out VPT2 calculations [22,23] for a series of 21 NH···O species (Table 2), comprising weak as well as strong hydrogen-bonding. However, as shown in Appendix A, the anharmonic NH stretching wavenumbers are essentially linearly related to the harmonic ones (R^2^ = 0.986, SD = 27 cm^−1^) and do not explain the non-linearity of the present correlations. On the other hand, the VPT2 anharmonic wavenumbers are numerically much closer to the experimental values than the harmonic ones, particularly for compounds with strong hydrogen bonds. This is in contrast to the results for OH···O systems, in which the VPT2 approximation significantly underestimated the OH stretching wavenumbers for strongly hydrogen-bonded systems (see above) [20,21,24,25]. Additional calculations using the B3PW91 functional [17,19] were performed for the compounds **1**, **6**, **16**, **17**, **23**, **29**, and **32**, covering a broad range of NH stretching wavenumbers. The computed harmonic wavenumbers are essentially linearly related to those obtained with B3LYP (R^2^ = 0.9999, SD = 2.6 cm^−1^).

### 3.3. Structures

The calculated molecular geometries evidently play important roles in the predicted NH stretching wavenumbers for the present NH···O linkages. An excellent linear correlation is observed for calculated wavenumbers and calculated NH bond lengths: R^2^ = 0.988, SD = 15 cm^−1^ (Appendix A). Much cruder correlations are found between wavenumbers and N····O and H···O distances. This is similar to the situation observed for OH···O systems [24].

Most of the investigated compounds have substituents with torsional degrees of freedom. In order to estimate the possible influence on the computed NH stretching wavenumber, calculations on several compounds were performed with consideration of different conformations. In general, the influence of the conformation is found to be of minor importance for the calculated wavenumber. For example, as shown in Appendix A, four conformational equilibrium geometries with different energies are located for **17**, but the same NH stretching wavenumber was calculated for all four structures.

The *N*-phenyl groups present in several of the investigated compounds were calculated to be twisted out of the plane of the –NH···O– moieties, and substituents at the phenyl ring were expected to have a minor effect on the NH stretching frequency. This is in agreement with the predicted and observed wavenumbers, except in the case of the observed values for **25** and **26** (see Section 4). Some of the enaminones and enaminoesters may occur both as *E* and *Z* forms, and the ratio between the two may depend on the solvent [39]. These and other points are discussed in the following section.

## 4. Discussion

NH stretching bands in compounds with weak NH···O hydrogen bonds are generally easy to identify (e.g., Figure 1), but in compounds with stronger hydrogen bonds, the band may be overlapped by CH stretching or other absorption bands. In those cases, deuteration is frequently a good tool. Provided an adequate conversion factor is known, the NH stretching wavenumber (ν_NH_) can be estimated from the observed ND band (ν_ND_). The isotope ratio ν_NH_/ν_ND_ was originally investigated by Novak [53] and more recently by Sobczyk et al. [54], showing a distinct dependence on the NH stretching frequency. However, in both cases, the majority of the data are from intermolecular hydrogen bonds. The intramolecular data reported by Sobczyk et al. [54] are for protonated dimethylaminonaphtalenes, DMANs, quite different from the compounds considered in this paper.

Figure 5 shows a plot of the measured ν_NH_/ν_ND_ ratios against ν_ND_ for a series of C-type esters with relatively weak hydrogen bonds. These results do not necessarily follow the trends described by Novak [53] and by Sobczyk et al. [54]. The average values of ν_NH_/ν_ND_ and ν_ND_ for these compounds are about 1.354 and 2400 cm^−1^, respectively. For the compounds of B type, such as **6, 8, 9, 10, 11**, **14**, **44**, and **45,** the observed values of ν_ND_ are lower (Table 3), amounting to an average close to 2265 cm^−1^. This indicates stronger hydrogen bonding, and according to the direction suggested by Novak [53], we expect a smaller effective value of the ν_NH_/ν_ND_ ratio. We shall tentatively assume the value ν_NH_/ν_ND_ = 1.34 for these compounds, leading to the ν_NH_ wavenumbers listed in Table 3. The wavenumbers obtained in this manner are in the range 2975–3063 cm^−1^ (they are not included in the correlation analyses). This prediction supports the assumption that these NH stretching bands are overlapped by absorption due to CH stretching transitions. The predicted wavenumbers are consistent with the observed NH chemical shifts close to 12 ppm (Table 2), indicating wavenumbers around 3100 cm^−1^ according to the correlation in Figure 2. They are also in good agreement with the semiempirical P(Harm) predictions based on Equations (1) and (2) (Table 2).

In order to establish useful correlations between observed and theoretical NH stretching wavenumbers ν_NH_, it is crucial to have reliable experimental assignments. Correlations between ν_NH_ and the NH proton chemical shift δ_NH_ have been demonstrated by Dudek [7] and by Gilli et al. [38]. A criterion for including data in the correlation analyses could therefore be that the experimental NH stretching wavenumbers should be consistent with the general correlation between experimental NH stretching frequencies and experimental NH chemical shifts. The plot of ν_NH_ wavenumbers vs. δ_NH_ chemical shifts is shown in Figure 2. The correlation is non-linear, in contrast to the linear relationship obtained for a range of related systems by Dudek [7]. Probably the extension of the range of compounds to include species of the types H, J, and L has led to the non-linear correlation. However, the points for **25** and **26** are striking outliers, and the wavenumbers for these compounds are not included in the correlation analyses. The δ_NH_ values reported [38] for these nitroso derivatives are exceptionally large (Table 2). The reason for the deviation from the correlation curve could be anisotropy effects, such contributions have recently been investigated in similar compounds [56]. Tautomerism could be another reason for a poor fit; deuterium isotope effects on ^13^C chemical shifts are good tests for tautomerism [57]. Derivatives of 4-amino-3-pentene-2-one (APO), linear and cyclic enaminones, and lactones have been shown to be non-tautomeric [39]. Some enaminones, esters, and lactones may exist both as *E* and *Z* forms; the assumed configurations of the investigated compounds are indicated in Table 1.

It is apparent that structure plays a great role in determining the NH stretching frequencies. The plot in Appendix A shows a very tight correlation between ν_NH_ and the calculated NH bond length. In general, the NH stretching wavenumbers will depend on the strength of the hydrogen bond. The correlation of observed wavenumbers with the hydrogen bond strengths estimated by the Espinosa method [58] leads to SD = 64–76 cm^−1^, depending on the choice of fitting function (Appendix A).

In the following, we discuss a number of literature data using the established correlation equations relating observed and calculated NH stretching wavenumbers (Equations (1) and (2)). A characteristic example is 4-(methylamino)pent-3-en-2-one (**12**). For the observed NH stretching wavenumber, Dudek [7] reported 3100 cm^−1^, while Raissi et al. [9] reported 3171 cm^−1^, and Kidwai et al. [8] assigned a feature at 3263 cm^−1^. As indicated by the P(Harm) values listed in Table 2, Equations (1) and (2) predict 3188 and 3183 cm^−1^ for **12**, pointing to the value 3171 cm^−1^ published by Raissi et al. [9].

Kidwai et al. [8] have published IR data for several β-enaminones. For (*Z*)-4-(phenylamino)pent-3-en-2-one (**6**) and (*Z*)-4-(*p*-nitrophenylamino)pent-3-en-2-one, they reported NH stretching wavenumbers equal to 3000 and 3362 cm^−1^, respectively. These assignments imply that the substitution of a nitro group in the *para* position of the *N*-phenyl ring of **6** shifts the NH stretching wavenumber by more than 300 cm^−1^. For the *p*-nitro derivative, we obtain with B3LYP/6-311++G(d,p) and B3LYP/6-31G(d) the harmonic wavenumbers 3229 and 3247 cm^−1^, respectively, leading to the P(Harm) predictions 3051 and 3063 cm^−1^. The present predictions indicate that the wavenumber 3362 cm^−1^ published by Kidwai et al. for (*Z*)-4-(*p*-nitrophenylamino)pent-3-en-2-one is much too large.

A number of other discrepancies are found in the literature. A large one is found for the compounds of D type (R = substituted phenyl), **23** and **24**. Gilli et al. [38] report NH stretching frequencies close to 2600 cm^−1^, whereas Benkheira and Amari [59] report 3390 cm^−1^. Benkheira and Amari discussed the possibility of a tautomeric equilibrium and concluded that the NH form is dominant, as is also assumed by Gilli et al. The NH chemical shifts are 15.6–15.9 ppm. Hence, the NH stretching frequency 3390 cm^−1^ reported by Benkheira and Amari seems much too high; the correlation in Figure 2 suggests wavenumbers around 2600 cm^−1^. This prediction is consistent with the semiempirical P(Harm) values (Table 2). Hence, the solid-state value reported by Gilli et al. [38] close to 2600 cm^−1^ seems realistic.

As mentioned above, the experimental wavenumbers for **25** and **26** are not included in the correlation analyses. The latter compound differs from the former only in the substitution of a methoxy group in the *meta* position of the *N*-phenyl ring, which is twisted out of the plane. Very similar NH stretching wavenumbers were computed and predicted for the two compounds (Table 2). However, the reported wavenumber for **26** is 220 cm^−1^ lower than that of **25 [38]**, a difference that seems hard to explain. Unfortunately, the IR spectra of the two compounds were not published.

Kolev and Angelov [60] investigated 1,1,1-trichloro-3-(1-phenethylamino-ethylidene)-pentane-2,4-dione (Figure 1, Type A: R_1_ = (C=O)CCl_3_, R_2_ = CH_3_, R_3_ = CH_2_CH_2_Ph, X = CH_3_). They assigned the observed IR spectrum to the *E* configuration and reported ν_NH_ = 3414 cm^−1^ in the solid state and δ_NH_ = 12.03 ppm. These values yield a point far from the correlation line in Figure 2. With δ_NH_ = 12.03 ppm, the correlation equation in Figure 2 predicts ν_NH_ = 3127 cm^−1^, indicating a relatively strong hydrogen bond. With B3LYP/6-311++G(d,p) and B3LYP/6-31G(d), we obtain harmonic NH stretching wavenumbers equal to 3272 and 3286 cm^−1^, respectively. With these values, Equations (1) and (2) predict 3106 and 3117 cm^−1^, consistent with the value indicated by correlation with the δ_NH_ value. Unfortunately, Kolev and Angelo [59] did not publish this region of the experimental spectrum.

An NH stretching frequency of 3317 cm^−1^ was recently reported for ethyl 2-(benzylamino)cyclopent-1-encarboxylate [36]. Our B3LYP/6-311++G(d,p) and B3LYP/6-31G(d) calculations yield 3451 and 3440 cm^−1^, respectively. This leads to P(Harm) values equal to 3271 and 3271 cm^−1^, in fair agreement with the experimental result.

The NH stretching wavenumbers for **27**, **30**, and **31** have apparently not been determined, but δ_NH_ values close to 13.3, 8.84, and 10.26 ppm have been published (Table 2) [7,39,40,41]. On the basis of the correlation in Figure 2, ν_NH_ values equal to 3007, 3295, and 3238 cm^−1^ can be estimated, respectively, in satisfactory consistency with the predicted P(Harm) values listed in Table 2.

In a number of papers [8,31,35,43,55], enaminones have been studied, but the NH stretching frequencies have not been assigned for all compounds. P(Harm) values predicted with Equations (1) and (2) may be of help in analyzing these spectra.

## 5. Conclusions

In this work, we considered NH stretching wavenumbers for a large number of species with intramolecular NH···O hydrogen bonding, ranging from weakly to strongly bonded systems. Additionally, a few systems with no intramolecular hydrogen bonding were included. The assignment of the NH stretching bands for a number of compounds was supported by measurements of the ND stretching wavenumbers of the deuterated species. The assignments were also supported by consideration of the correlation between observed NH stretching wavenumber and NH proton chemical shift. This correlation was well described by an exponential function (SD = 34 cm^−1^), in contrast to the linear relationships found in the literature for related systems.

Excellent exponential relationships were also established between observed NH stretching wavenumbers and harmonic wavenumbers predicted by B3LYP/6-31G(d) and B3LYP/6-311++G(d,p) calculations. With SD = 38 cm^−1^, these semiempirical correlation equations should be of considerable predictive value, as demonstrated in the Discussion Section of this article (Section 4). This result is significant because rigorous theoretical procedures exceeding the harmonic approximation, such as the VPT2 procedure, tend to be impractical for large molecules, requiring orders of magnitude more computing time than the harmonic analysis.

The non-linearity of these relationships is in contrast to previous results for OH···O systems, which were characterized by corresponding linear relationships. We have presently no obvious explanation for the apparent difference between NH···O and OH···O systems. NH stretching wavenumbers computed with the anharmonic VPT2 approximation are found to be linearly related to the corresponding harmonic values, offering no explanation of the non-linearity of the present correlations. It would seem that more advanced theoretical calculations, beyond the VPT2 approximation, are called for.

## Data Availability

Not applicable.

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
