# Peer review of "NH Stretching Frequencies of Intramolecularly Hydrogen-Bonded Systems: An Experimental and Theoretical Study"

_molecules, 2021, doi:10.3390/molecules26247651_

Round 1

Reviewer 1 Report

The manuscript described ''The vibrational NH stretching transitions in secondary amines with intramolecular NH···O hydrogen bonds were investigated by experimental and theoretical methods, considering a large number of compounds and covering a wide range of stretching wavenumbers''  but the paper needs very significant improvement before acceptance for publication. There are some questions about the manuscript as follows:
1. There are some errors in spelling, syntax, punctuation, usage of capital letters, consistency in language style in the manuscript that must be improved.
2. Full characterization of all synthesized compounds is required in the experimental section.

3. All the 1HNMR and 13CNMR must be attached as supplementary information.

4. Scheme 1 must be redrawn using ACS document 1996 and all structures cleaned up.

5. The cell values do not align please correct them.

6. Use 4 significant figures for the RNH values in table 2

7. It would be added value if these NH···O hydrogen bonds were also investigated in the solid state using XRD analysis.

8. Check the font of the reference numbers in the text and correct

Author Response

Reviewer 1
The manuscript described ''The vibrational NH stretching transitions in secondary amines with intramolecular NH···O hydrogen bonds were investigated by experimental and theoretical methods, considering a large number of compounds and covering a wide range of stretching wavenumbers''  but the paper needs very significant improvement before acceptance for publication. There are some questions about the manuscript as follows:

1. There are some errors in spelling, syntax, punctuation, usage of capital letters, consistency in language style in the manuscript that must be improved.
      RESPONSE:  We have attempted to improve the language and hope we have found all significant errors.

2. Full characterization of all synthesized compounds is required in the experimental section.
      RESPONSE: The characterization of the compounds is provided together with the NMR spectra as Suppl. Mat. S21.

3. All the 1HNMR and 13CNMR must be attached as supplementary information.
      RESPONSE: Spectra have been included in Suppl. Mat. S21.

4. Scheme 1 must be redrawn using ACS document 1996 and all structures cleaned up.
      RESPONSE: A new Scheme 1 has been produced with the recommended format. The molecular structures in Figures 1 and 3 have been modified accordingly.

5. The cell values do not align please correct them.
      RESPONSE: The misalignment is a result of the editorial procedure. We have tried to improve the alignment.

6. Use 4 significant figures for the RNH values in table 2.
      RESPONSE: Done.

7. It would be added value if these NH···O hydrogen bonds were also investigated in the solid state using XRD analysis.
      RESPONSE: We appreciate the suggestion. However, we do not think we have time to perform a proper analysis of XRD data, considering solid state effects, crystal packing, etc.

8. Check the font of the reference numbers in the text and correct
      RESPONSE: In the submitted MSWord document the font was Times New Roman for text and reference numbers. Something has happened during the editorial procedure when changing to Palatino Linotype. Not all reference numbers were changed correctly. We have tried to find and correct the errors. 

Reviewer 2 Report

In this article, the authors describe the spectral characteristics of NH in the hydrogen bridge NH...O. It is a continuation of similar OH...O binding studies performed 10 years ago. The rationale for undertaking the research is a bit puzzling. I am curious what prompted the Authors to deal with N-H...O bonding in place of the earlier O-H...O. The explanation should be cleared.
Anyway, I think the article has some educational significance and could be published after few major and minor points are taken into account:       

Major:

1) The authors should give the reason for the nonlinear v(NH) vs delta correlation as it was (according to Dudek) linear for OH-O.

2) Performing calculations with the Grimme dispersion correction did not make sense here, as D3 is important for intermolecular interactions, especially stacking. Here, calculations are performed for isolated molecules. Rather, the author could check B3PW91 (it often performs better than B3LYP) or any newer suitable functional.

Minor:

183: should be SD = 34.3 cm-1
186: "are reasonable": Taking into account that (i) this fit is rather unphysical
(ii) moving the points by say 50cm-1 one way or the other would also give rather good fit, this phrase is rather an empty cliche.
Figure 5: Why is the y-axis reversed?

Author Response

Reviewer 2
In this article, the authors describe the spectral characteristics of NH in the hydrogen bridge NH...O. It is a continuation of similar OH...O binding studies performed 10 years ago. The rationale for undertaking the research is a bit puzzling. I am curious what prompted the Authors to deal with N-H...O bonding in place of the earlier O-H...O. The explanation should be cleared.
      RESPONSE: The background for the investigation is described in the Introduction section. As indicated by the Reviewer, the present study is an extension of our previous OH···O binding studies. We have for some time wished to investigate the corresponding NH···O systems. The NH···O type of linkage has the advantage that a wide range of compounds can be investigated; in general, systems of the type NH···O=C with an intervening double bond are not tautomeric, in contrast to the corresponding OH···O=C systems. We have added a remark on this aspect to the Introduction section. - The ‘delay’ of 10 years is due to technical and personal factors of no relevance to the reported results. 

Anyway, I think the article has some educational significance and could be published after few major and minor points are taken into account:       

Major:
1) The authors should give the reason for the nonlinear v(NH) vs delta correlation as it was (according to Dudek) linear for OH-O.
      RESPONSE: We are aware of the controversial nature of this result. We also expected a linear relationship similar to the one observed by Dudek. However, we have in this work extended the range of compounds to include species of types J, K, and L, apparently resulting in the observed non-linear relationship. The correlation in Fig. 2 involves only experimental values, most of them from the literature, and we have no reason to doubt their reliability. The exponential regression yields R2 = 0.975, SD = 34.3 cm–1, while the corresponding linear regression gives R2 = 0.837, SD = 93.3 cm–1. Hence, the indication of a non-linear correlation is evident. We have added the sentence "Probably the extension of the range of compounds to include species of the types J, K, and L has led to the non-linear correlation” to the first paragraph on page 13.

2) Performing calculations with the Grimme dispersion correction did not make sense here, as D3 is important for intermolecular interactions, especially stacking. Here, calculations are performed for isolated molecules. Rather, the author could check B3PW91 (it often performs better than B3LYP) or any newer suitable functional.
      RESPONSE: Our attention was drawn to Ref. [54] where Grimme et al. state that "The [DFT-D3] method has been assessed on standard benchmark sets for inter- and intramolecular noncovalent interactions". We thus found it relevant to try out B3LYP-D3 for the present study of "intramolecular noncovalent interactions". But the Reviewer finds that these calculations "make no sense here", so we have removed these results from the manuscript. 
      The Reviewer suggests that we check the functional B3PW91 (in the Introduction we actually mention B3PW91 as one of the two functionals recommended by Scott and Radom). We have performed additional B3PW91/6-311++G(d,p) calculations on a series of seven compounds covering a wide range of NH stretching wavenumbers. The computed harmonic wavenumbers are essentially linearly related to those obtained with B3LYP (R2 = 0.9999, SD = 2.6 cm–1). We have added a remark on this result to the last paragraph of Section 3.2.   

Minor:
183: should be SD = 34.3 cm-1
      RESPONSE: Corrected!

186: "are reasonable": Taking into account that (i) this fit is rather unphysical (ii) moving the points by say 50 cm-1 one way or the other would also give rather good fit, this phrase is rather an empty cliche.
      RESPONSE: The sentence with "are reasonable" has been removed.

Figure 5: Why is the y-axis reversed?
      RESPONSE: The orientation of the axes was chosen according to the trend reported by Novak [56]: Decreasing NH stretching wavenumber corresponds to decreasing isotope ratio. 

Round 2

Reviewer 1 Report

  1. In the experimental section of the 13C NMR it should be 125 MHz instead of 126 MHz.
  2. Correct subscripts and superscripts in the experimental also check the spacings to be consistent.

Author Response

Reviewer 1:

Comments and Suggestions for Authors
1.   In the experimental section of the 13C NMR it should be 125 MHz instead of 126 MHz.
2.   Correct subscripts and superscripts in the experimental also check the spacings to be consistent.

This concerns the experimental section, S21, in the Supplementary Material document. We have scrutinized the text and corrected the errors. We have also corrected two typos in the main manuscript.